# ACTH(6–9)PGP Peptide Protects SH-SY5Y Cells from H_2_O_2_, *tert*-Butyl Hydroperoxide, and Cyanide Cytotoxicity via Stimulation of Proliferation and Induction of Prosurvival-Related Genes

**DOI:** 10.3390/molecules26071878

**Published:** 2021-03-26

**Authors:** Mikhail G. Akimov, Elena V. Fomina-Ageeva, Polina V. Dudina, Ludmila A. Andreeva, Nikolay F. Myasoyedov, Vladimir V. Bezuglov

**Affiliations:** 1Shemyakin-Ovchinnikov Institute of Bioorganic Chemistry, Russian Academy of Sciences, Miklukho-Maklaya 16/10, 117997 Moscow, Russia; evfa56@gmail.com (E.V.F.-A.); polinadudkinz@gmail.com (P.V.D.); vvbez@ibch.ru (V.V.B.); 2Institute of Molecular Genetics of National Research Centre, Kurchatov Institute, Ploshchad’ Akademika Kurchatova 2, 123182 Moscow, Russia; landr@img.ras.ru (L.A.A.); nfm@img.ras.ru (N.F.M.)

**Keywords:** ACTH(6–9), neuroprotection, MPP^+^, H_2_O_2_, *tert*-butyl hydroperoxide, cyanide, melanocortins, oxidative stress

## Abstract

Stabilized melanocortin analog peptide ACTH(6–9)PGP (HFRWPGP) possesses a wide range of neuroprotective activities. However, its mechanism of action remains poorly understood. In this paper, we present a study of the proproliferative and cytoprotective activity of the adrenocorticotropic hormone fragment 6–9 (HFRW) linked with the peptide prolyine–glycyl–proline on the SH-SY5Y cells in the model of oxidative stress-related toxicity. The peptide dose-dependently protected cells from H_2_O_2_, *tert*-butyl hydroperoxide, and KCN and demonstrated proproliferative activity. The mechanism of its action was the modulation of proliferation-related *NF-κB* genes and stimulation of prosurvival *NRF2*-gene-related pathway, as well as a decrease in apoptosis.

## 1. Introduction

One of the actively studied classes of endogenous peptide regulators is adrenocorticotropic hormone (ACTH)/melanocyte-stimulating hormone (MSH)-like peptides, which are currently combined under the term melanocortins (MCs). Studies of the effects of these peptides over the past 40 years showed that the spectrum of the physiological activity of these peptides is very wide. MCs are involved in the regulation of memory and attention, emotional status, sexual and eating behavior, pain sensitivity, and some other physiological functions [1]. The MC family includes ACTH and MSH (α-, β-, and γ-MSH), as well as fragments of these hormones and their synthetic analogs. Of particular interest is the ability of ACTH and MSH, as well as their fragments lacking hormonal activity, to accelerate the training of animals and improve the preservation of skills. The most active of the studied peptides is ACTH(4–10). The smallest ACTH fragment retaining nootropic activity is ACTH(4–7). However, it was also shown that the ACTH(7–10) fragment has insignificant nootropic activity. Elongation of the molecule to ACTH(7–16) leads to an increase in activity to a level comparable to ACTH(4–7) and ACTH(4–10) [2,3].

The described wide range of physiological activity of melanocortins opens up opportunities for the use of drugs of this class in clinical practice in various pathological conditions. However, an obstacle to the use of these compounds in the clinic is low bioavailability and insufficient action duration. The duration of the neurotropic effects of the natural ACTH(4–10) fragment is 30–60 min [4]. In the organs and tissues of most warm-blooded animals, peptides are usually hydrolyzed by widely represented exo- and endopeptidases with low specificity. However, they do not cleave the AA–Pro bonds, where AA is any amino acid, as well as those of other sequences enriched in proline residues [5]. Specific prolyl hydroxylases are concentrated mainly in individual organs and tissues in a small amount. Therefore, the prolongation of the effects of the peptide and increase in its peptidase resistance was found to be possible by including sites enriched in proline residues in the structure of the molecule. A method for stabilizing synthetic peptides has previously been reported using proline motifs found in natural proteins to prevent the action of peptidases [6]. Based on this, it was suggested that the addition of a proline-enriched chain to the C-terminus of the ACTH fragments would lead to both an enhancement of the effect of the peptide and its prolongation.

One of such proline-enriched chains is the collagen-derived matrikine tripeptide proline–glycine–proline (PGP), which was described as a neutrophil chemoattractant. PGP functions as a primitive and conserved damage-associated molecular pattern generated during infection or injury and subsequently acts to shape ensuing inflammatory and repair processes. PGP also possesses a profound capacity to promote proliferation, radial spreading, and prominent lamellipodia formation in human lung bronchial epithelial cells [7].

PGP was successfully used for the stabilization of various nootropic peptides [8], including ACTH fragments. For example, the peptide drug Semax (MEHFPGP) is composed of an ACTH(4–7) fragment that is flanked at the C-terminus with PGP peptide. The duration of the physiological effect of Semax in the T-shape maze educational test was 50 times that of the natural ACTH(4–10) peptide [9]. The corresponding fusion peptide ACTH(6–9)PGP (HFRWPGP) was obtained by conventional peptide synthesis [10], and its stability towards various peptidases was investigated. Under all experimental conditions, the bond between the tryptophan and proline (HFRW–PGP) in the peptide was resistant to hydrolytic splitting [10], demonstrating the correctness of the mentioned strategy for the peptide activity prolongation.

The common sequence for all MCs is ACTH(6–9) or HFRW, which is necessary for binding to all known types of MC receptors [11]. Animal experiments demonstrated that ACTH(6–9)PGP (HFRWPGP) exhibited nootropic and anxiolytic activities. Intraperitoneal administration of ACTH(6–9)PGP helps to reduce high anxiety levels in animals both with aggressive and submissive types of behavior under conditions of experimental social stress via their psychomodulatory effects [12]. The peptide also exhibits a psychomodulatory effect and relieves symptoms of anxiety–depressive disorders caused by exposure to social stress [13]. In addition, ACTH(6–9)PGP enhanced the viability of cultured cortical neurons under glutamate excitotoxicity [14].

Despite a profound biological activity, the mechanisms of ACTH(6–9)PGP neuroprotective activity remain poorly understood.

In this paper, we show for the first time that the modified adrenocorticotropic hormone fragment ACTH(6–9)PGP has neuroprotective activity in the H_2_O_2_, *tert*-butyl hydroperoxide, and KCN toxicity setting that models cell-death-inducing mechanisms in Parkinson’s disease and ischemic stroke. We demonstrate that the neuroprotective activity of the peptide is realized via proliferation increase by modulation of proliferation-related *NF-κB* genes and stimulation of prosurvival *NRF2*-gene-related pathway, as well as a decrease in apoptosis.

## 2. Results

### 2.1. ACTH(6–9)PGP Stimulated Cell Proliferation and Increased Cell Survival after H_2_O_2_, tert-Butyl Hydroperoxide, MPP^+^, and KCN Treatment

To study the neuroprotective activity of ACTH(6–9)PGP, four oxidative-stress-related models were used: oxidative stress induction by H_2_O_2_, *tert*-butyl hydroperoxide (tBH), KCN, and MPP^+^ cytotoxicity. The cytotoxicity of H_2_O_2_ (EC_50_ = 475 µM) and MPP^+^ (EC_50_ = 1.3 mM) was determined beforehand [15], and the cytotoxicity of KCN and tBH was explicitly measured (Figure 1); EC_50_ was 90.6 µm (C.I. 87.05 to 94.16 µM) for KCN and 27.5 µM (C.I. 24.52 to 30.97 µM) for tBH.

In the experiments with the peptide, the added toxic agent concentration was chosen to induce 50 to 70% cell death in 24 h; the peptide was added together with the toxin.

ACTH(6–9)PGP increased cell viability under treatment with H_2_O_2_, tBH, and KCN, but not with MPP^+^ (Figure 2A–D). For H_2_O_2_ and tBH, the protective activity increased until the maximum at the peptide concentration around 1–10 µM (Figure 2A,B) and then decreased. For KCN, there was a dose-dependent increase in cell survival with a maximum at 100 µM (Figure 2C).

The effect in the tBH model was more pronounced than in the H_2_O_2_ one. However, H_2_O_2_ is produced in vivo during oxidative stress [16,17] and thus represents a more physiological setting. Therefore, further studies on the ACTH(6–9)PGP action mechanism were conducted in the H_2_O_2_ model.

### 2.2. ACTH(6–9)PGP Decreased Apoptosis and Increased Cell Viability but Did Not Affect Acute ROS Level

Three possible mechanisms of the protective action of ACTH(6–9)PGP could be proposed: proliferation stimulation, apoptosis inhibition, and ROS level decrease either as a direct ROS scavenging effect or via the activation of appropriate cellular enzyme systems. The latter hypothesis is possible, as all the toxic agents used are linked to reactive oxygen species generation [18,19].

To test the first two hypotheses, we studied SH-SY5Y proliferation increase after the ACTH(6–9)PGP treatment and apoptosis levels after the combined application of the peptide and H_2_O_2_. To evaluate cell proliferation, MTT assay and BrdU incorporation assay were used. After the peptide treatment, both MTT staining and BrdU incorporation increased compared to the untreated control, and thus the peptide stimulated cell proliferation (Figure 3A,B). H_2_O_2_ treatment increased the fluorescence of the phosphatidylserine sensor, indicating the induction of apoptosis; this fluorescence in the peptide-treated cells decreased compared to the control without the peptide, and thus this treatment decreased apoptosis, but the effect was statistically significant only at 100 μM of the peptide (Figure 3A). Finally, we used the DCFH-DA dye to detect ROS levels. Its fluorescence increased after the H_2_O_2_ treatment, indicating the accumulation of ROS, but did not change after the ACTH(6–9)PGP treatment. Thus, no ROS scavenging effect was observed after a 1-h incubation (Figure 3C).

### 2.3. ACTH(6–9)PGP Protection against KCN Cytotoxicity Is Inhibited by the MEK, PKC, PLC, and Ras Inhibitors

To evaluate the signaling pathways involved in the peptide protective activity, we used a panel of receptor and intracellular signal transduction component inhibitors. As extracellular ROS-generating oxidative stress inducers like H_2_O_2_ can oxidize these inhibitors, we used the KCN-based toxicity model. Of the inhibitors tested, only those for the protein kinases C and A (PKC + PKA), phospholipase C (PLC), mitogen-activated protein kinases 1 and 2 (MEK1,2), and Ras removed the protective effect of the peptide (Figure 4), and thus these components participate in the peptide action. Overall, the observed inhibitor response points to the activation of the PLC->PKC->Ras->MEK signal transduction pathway [20,21,22].

### 2.4. ACTH(6–9)PGP Did Not Alter Cellular cAMP Content

Considering that the receptor for the full-length ACTH is a GPCR coupled to a G_αs_ subunit, which in turn activates cAMP synthesis, and that one of the active inhibitors was the mixed PKA/PKC one, we decided to check whether the action of ACTH(6–9)PGP affects the concentration of this second messenger. We analyzed cellular cAMP content after a 20-min incubation with the peptide, while prostaglandin PGE_2_ was used as a positive control (Figure 5). We detected neither increase nor decrease in the cAMP concentration, and thus the participating kinase is PKC and not a cAMP-dependent PKA.

### 2.5. ACTH(6–9)PGP Decreased the Expression of the NF-κB- and Nrf-2-Related Genes but Not of the Antioxidant Enzymes

The downstream mechanism for the protection against ROS cytotoxicity could be the induction of the cellular antioxidant genes.

To check this possibility and to further elaborate the proproliferative activity of the peptide, we analyzed the mRNA levels of a set of signaling pathways after ACTH(6–9)PGP application both alone and in combination with H_2_O_2_ for 24 h [23]:NF-κB pathway: *AKT3*, *IκB*, and *NF-κB*;Nrf-2 pathway: *Nrf2*, *HO-1*, *GST*, *NQO1*, *GCLC*, *SOD1*, *SOD2*, and *CAT* (catalase);MAPK pathway: *JNK*, *P38*, *MKP1*, *PP2A*, *PP5*, and *Ki-67*;DNA-damage-related pathway: *P53*.

The mRNA levels of the *AKT3*, *CAT*, *SOD1,* and *PP2A* genes were below the nonspecific amplification (no template control) threshold (data not illustrated). Peptide treatment alone activated NF-κB pathway expression, stimulated the *P38* protein kinase, and activated the expression of several Nrf2 pathway components (Figure 6). In combination with H_2_O_2_, the stimulation of the NF-κB pathway was somewhat decreased, and a small decrease in the *Ki-67* gene expression was also observed. *P38* expression was further enhanced. Two specific gene expression changes were observed for the combination of the peptide with H_2_O_2_: first, the addition of ACTH(6–9)PGP restored the levels of *P53* and *PP5* to the control level, and second, the expression of the *GCLC* gene was increased. These changes agree with the prosurvival and proproliferative action of the peptide.

## 3. Discussion

The recent data on neuromodulatory and neuroprotective properties of the ACTH(6–9)PGP peptide [12,13,14] clearly showed promising perspectives of this molecule as a new pharmaceutical. The aim of this work was to identify the intracellular systems that participate in the neuroprotective activity of the ACTH(6–9)PGP peptide. Such data could give an insight into the possible interactions with other neuroprotective drugs to produce a drug combination with enhanced activity in the future. The obtained results point to the following two peptide properties: (1) the peptide evokes a proproliferative effect; (2) the peptide stimulates the NF-κB and Nrf2 signaling pathways.

The peptide enhanced cell viability in the H_2_O_2_, tBH, and KCN cytotoxicity models but not in the MPP^+^ cytotoxicity model. The ability of the peptide to enhance cell viability increased in the sequence H_2_O_2_ < tBH ≤ KCN.

The increase in cell survival in the H_2_O_2_ model after the ACTH(6–9)PGP treatment was comparable to the peptide’s proproliferative effect. In this model, to induce cell death, H_2_O_2_ is added to the cells in vast quantities. Therefore, the antioxidant systems of the cell are hardly able to cope with the ROS from H_2_O_2_, even if they are stimulated by the peptide, and the net increase in the cell viability after ACTH(6–9)PGP treatment should be due to the proliferation increase. The cytotoxic concentration of tBH was more than 20 times lower, and thus it is quite expectable that the peptide was able to protect many more cells in this model. The observed cytoprotective activity of the peptide was comparable to that of the selenium-rich peptide fraction from selenium-rich yeast protein hydrolysate [24] and that of the mitochondria-targeted peptide SS31 [25].

The effect of the peptide in the KCN cytotoxicity model was somewhat different from H_2_O_2_ and tBH, as it was active only at quite high concentrations. This discrepancy could be explained by the fact that tBH and H_2_O_2_, due to their short lifetime in the culture medium, produce a relatively short stress [26], while KCN, which targets mitochondria [27], produces a long-term ROS production and thus may require a much higher activation of the antioxidant systems.

MPP^+^, a neurotoxin that plays dominant neurotoxic roles in selectively damaging catecholaminergic neurons, including dopaminergic neurons, has been widely used in the experimental models of PD, and it can operate in extracellular or intracellular oxidation, yielding ROS that lead to toxic downstream molecules and result in neuronal damage [28]. It was demonstrated that the treatment of SH-SY5Y cells with MPP^+^ results in a significant increase in ROS concentration [18]. However, MPP^+^ is quite unstable, and thus it causes only a temporary ROS concentration increase. In this condition, the antioxidant activity of a substance should play a substantial role, as was shown, for example, for α-lipoic acid [29]. ACTH(6–9)PGP failed to decrease the ROS concentration after an hour and thus does not act as a direct ROS scavenger. Its structure (His–Phe–Arg–Trp–Pro–Gly–Pro) does not entirely preclude such activity, as phenylalanine and tryptophan residues may react with ROS [30]. In this assay, the peptide appeared to be inferior to the redoxin-mimetic peptide PSELT [31] and glucagon-like peptide-2 [32]. It could be proposed that the concentration of the peptide is not enough for this activity to manifest. The vast quantities of MPP^+^ required to induce cytotoxicity also make it hardly possible for the cellular antioxidant machinery to be able to cope with such stress, and this is the most probable explanation for the lack of peptide activity in this model.

The signaling behind the activity of ACTH(6–9)PGP requires further investigation; however, several hypotheses could be put forward to explain it. As the active peptide concentrations were much smaller than the toxic agent’s ones, ACTH(6–9)PGP should interact with some cellular signaling machinery rather than inactivate the added toxins or produced ROS directly. This agrees with the observed disappearance of the peptide’s protective action after the inhibition of the PLC->PKC->Ras->MEK signal transduction pathway. The participation of this signaling pathway is similar to the PACAP neuropeptide. PACAP was shown to have neuroprotective effects in PD models, but its complex pharmacological actions, as well as the short half-life, limit its clinical application [33]. ACTH(6–9)PGP is expected to be more stable, but the lack of data on its receptors makes it hard to predict its side effects.

Downstream of these signaling components, after the peptide treatment, the mRNA expression changed according to several patterns.

Peptide treatment alone activated *NF-κB* expression and decreased the expression of its inhibitor *IκB*; the latter change was also observed in the presence of H_2_O_2_. This change should activate the proproliferative NF-κB pathway [34,35] and agrees with the observed proproliferative action of the peptide. The increase in the *NF-κB* expression after the peptide treatment also agrees with the detected inhibitor activity, as this gene activity is activated by Ras [36].

ACTH(6–9)PGP treatment also stimulated the expression of the *P38* protein kinase and several Nrf2 pathway components. In the presence of H_2_O_2_, *P38* expression was further enhanced. PKC, which was one of the components of the detected peptide signaling, is also known to stimulate the activity of the Nrf2 transcription factor [37], and thus the observed activation of the expression of the Nrf2 targets *NQO1* and *HO-1* after the peptide treatment seems quite logical.

Three specific gene expression changes were observed for the combination of the peptide with H_2_O_2_: first, the addition of ACTH(6–9)PGP restored the levels of *P53* and *PP5* to the control level; second, the expression of the *GCLC* gene was increased; and finally, a small decrease in the *Ki-67* gene expression was observed. These changes mostly agree with the prosurvival and proproliferative action of the peptide, with some shift toward the former one in the presence of cytotoxic agents. The activation of the gene of the rate-limiting enzyme in the glutathione biosynthesis *GCLC* [38] is of particular interest, as it presents an interesting way of long-term defense against the oxidative stress induced by the peptide.

The direct target of the peptide is not clear. Based on the fact that the peptide represents a part of the adrenocorticotropic hormone, its receptor could be the first candidate. ACTHR is coupled to a G_αs_ subunit [39], and so its activation should lead to an increase in cAMP concentration. However, cAMP levels did not change after ACTH(6–9)PGP application (Figure 5), and thus ACTHR could be excluded from the list of potential targets.

Recently deorphanized cannabinoid receptors GPR18 and GPR55 are known to stimulate proliferation [40,41]; recently, a possibility of GPR55 peptide modulation was demonstrated [42]. In our experimental setting, the inhibitors of these two receptors (ML-193 at 2 µM and PSB CB5 at 0.5 µM) prevented ACTH(6–9)PGP protection against KCN cytotoxicity (Appendix A). Thus, these receptors could be at least allosteric targets for this peptide; however, a more detailed study of this interaction is out of the scope of this paper.

Other possibilities for the ACTH(6–9)PGP target include receptors like GLP1R, FGFR, and heat shock proteins. GLP1R is the receptor for the neuroprotective action of the peptide Glp-1 [43]. This receptor signaling is biased [44], so it could interact with ACTH(6–9)PGP. However, it is also coupled to G_αs_ and thus does not fit the observed data. FGFR activation by a peptide representing the receptor-binding domain of bFGF was shown to elicit a neuroprotective response in SH-SY5Y cells [45]. FGFR is a receptor tyrosine kinase, and it is linked to the Akt signal transduction pathway [46]. Therefore, this or a similar receptor could be the ACTH(6–9)PGP target. Heat shock proteins interact with other bioactive peptides [47] and increase cell proliferation [48], and this pathway should be kept in mind. A more detailed investigation of discussed receptor targets is necessary to elucidate mechanisms of cytoprotective activity of modified ACTH(6–9) peptide.

## 4. Materials and Methods

### 4.1. Materials

l-Glutamine, fetal bovine serum, penicillin, streptomycin, amphotericin B, Hanks’ salts, trypsin, DMEM, and (4,5-dimethylthiazol-2-yl)-2,5-diphenyltetrazolium bromide (MTT) were from PanEco, Moscow, Russia. 2′,7′-Dichlorodihydrofluorescein diacetate (DCFH-DA), HEPES, DMSO, d-glucose, MPP^+^, KCN, MPP^+^, l-NAME, *tert*-butyl hydroperoxide, and bovine serum albumin were from Sigma-Aldrich, St. Louis, MO, USA. 666-11, SB 202190, SP 600125, salirasib, FIPI, KN-93, ML-193, PSB C5, HA-1004, U-0126, KT-5720, and U-73 were from Tocris Bioscience, Bristol, UK. Total RNA Purification kit was from Jena Biosciences, Jena, Germany. MMLV reverse transcription kit and qPCR master mix qPCRmix-HS SYBR were from Evrogen, Moscow, Russia. cAMP determination kit and BrdU cell proliferation assay kit were from Abcam, Cambridge, MA, USA. DNase I was from Thermo Fisher Scientific, Waltham, MA, USA.

The peptide was synthesized by methods of classical peptide chemistry in solution using both protected and free L-amino acids, as described earlier [49]. Purity of the final compound was not less 98% (HPLC analysis, Appendix A).

### 4.2. Cell Culture

SH-SY5Y cells (ATCC CRL-2266) were maintained in the 1:1 MEM/F12 medium supplemented with 10% fetal bovine serum, 2 mM l-glutamine, 0.5 mM sodium pyruvate, 0.5% nonessential amino acids, 100 U/mL penicillin, 100 μg/mL streptomycin, and 2.5 μg/mL amphotericin B in a CO_2_ incubator with the atmosphere of 5% CO_2_ and 95% humidity at 37 °C. Cells were passaged every 72 to 96 h by washing with Versene’s solution and treatment with 0.25% trypsin with 0.53 mM EDTA in Hanks’ balanced salt solution.

### 4.3. Oxidative Stress Induction

For cell viability experiments, the cells were seeded at a density of 15,000 per well of 96-well plate in 100 μL of culture medium and incubated for 12 h. After that, the substance solution with or without the toxic agent in 100 μL fresh cell culture medium was added to the medium present in the wells and incubated for 24 h. Cytotoxicity was induced by either 475 μM of H_2_O_2_, 850 µM KCN, 1.3 mM MPP^+^ (freshly prepared1), or 15 µM *tert*-butyl hydroperoxide (all with the addition of 10 mM of HEPES, pH 7.4).

### 4.4. Cell Viability Assay

Cell viability was analyzed using the MTT test [50]. In short, the culture medium was removed from the wells, and 75 μL of the 0.5 mg/mL solution of MTT with 1 g/L D-glucose in Earle’s salts was added to each well and incubated for 90 min in the CO_2_ incubator at 37 °C. After that, 75 μL of 0.04 M HCl in isopropanol was added to the MTT solution in each well and incubated on a plate shaker at 37 °C for 30 min. The optical density of the solution was determined using a plate reader (Efos 9226, MZ Sapphire, Russia) at the wavelength of 570 nm with a reference wavelength of 620 nm.

### 4.5. cAMP Assay

cAMP levels were measured using a cAMP determination kit according to the manufacturer’s instructions. The cells were seeded at a density of 60,000 per well of 96-well plate and grown for 12 h. The substances were added in 100 μL fresh cell culture medium to the medium present in the wells and incubated for 20 min. After that, the cell culture medium was removed, and the cells were subjected to cAMP determination.

### 4.6. ROS Assay

ROS generation was measured using the DCFH-DA dye. The cells were seeded at a density of 60,000 per well of 96-well plate and grown for 12 h. After that, the medium was replaced with a fresh one with 25 μM of the dye, and the cells were incubated in the CO_2_ incubator at 37 °C for 1 h. After the incubation, the cells were washed twice with the culture medium and treated with the substances in the culture medium for 1 h at 25 °C. Cells treated with medium without H_2_O_2_ and substances were used as a control. After the incubation, the cells were washed twice with Hanks’ balanced salt solution with 25 mM HEPES and 1 mg/mL fatty acid-free bovine serum albumin, pH 7.4, and the fluorescence was measured using the plate reader Hidex Sense Beta Plus (Hidex, Turku, Finland), λ_ex_ = 490 nm, λ_em_ = 535 nm.

### 4.7. Apoptosis Assay

Apoptosis level was analyzed using the Apoptosis/Necrosis detection kit (ab176749, Abcam, Cambridge, UK). The cells were seeded at a density of 15,000 per well of 96-well plate and grown for 12 h. After that, 475 μM of H_2_O_2_ alone or with the peptide was added in 100 μL of the fresh medium to 100 μL of the old medium in the wells and incubated for 1 h at 37 °C in a CO_2_ incubator. After that, the medium was removed, and the cells were stained according to the manufacturer’s instructions using the phosphatidylserine sensor (apoptotic cells, green fluorescence), membrane-impermeable dye 7-AAD (necrotic cells, red fluorescence), and a live cell cytoplasm dye CytoCalcein Violet 450. The stained cells were photographed using an inverted fluorescent microscope Nikon Ti-S using a Semrock GFP-3035D filter cube with magnification 100×. For each well, 5 nonintersecting view fields were captured, and apoptotic cells were counted.

### 4.8. mRNA Assay

mRNA levels were analyzed using RT-qPCR. The cells were seeded at the density of 240,000 per well of a 24-well plate in 200 μL of culture medium and incubated for 12 h. The substances were added in 200 μL of fresh culture medium to the medium in the wells and incubated for 24 h. Total RNA was extracted using a Total RNA isolation kit according to the manufacturer’s protocol. The isolated RNA was treated with DNase I according to the manufacturer’s protocol. cDNA was synthesized using an oligo-dT primer using the MMLV reverse transcription kit. qPCR was performed using a SYBR Green containing master mix qPCRmix-HS SYBR with the following amplification protocol: 95 °C for 2 min, cyclic 95 °C for 10 s, 57 °C for 20 s, 72 °C for 15 s for 40 cycles using a Bio-Rad C1000 thermal cycler (Bio-Rad, Hercules, CA, USA). After the amplification, PCR product melting curve was recorded in the range from 65 to 95 °C. Primer sequences were generated using the IDT PrimerQuest service (https://eu.idtdna.com/PrimerQuest, accessed on 26 March 2021) and validated using the NCBI PrimerBLAST service [51], or they were taken from the paper by Jaafru et al. [23]. The primer sequences were as follows (5′-3′):

AKT3 forward AGGTGACACTATAGAATAAGACATTAAATTTCCTCGAA, reverse GTACGACTCACTATAGGGAATCCTCATCATATTTTTCAGGT;

Beta-2 microglobulin forward CAGCAAGGACTGGTCTTTCTAT, reverse ACATGTCTCGATCCCACTTAAC;

Catalase forward AGGTGACACTATAGAATAAGAAATCCTCAGACACATCT, reverse GTACGACTCACTATAGGGAATGTCATGACCTGGATGTAA;

GST forward AGGTGACACTATAGAATAATACATGGCAAATGACTTAAA, reverse GTACGACTCACTATAGGGATGATGTCTTCATTCCTTGAC;

GCLC forward AGGTGACACTATAGAATAATGAAGCAATAAACAAGCAC, reverse GTACGACTCACTATAGGGATGGAATGTCACCTGGAG;

GAPDH forward GAATGGGAAGCTGGTCATCAA, reverse CCAGTAGACTCCACGACATACT;

HO-1 forward AGGTGACACTATAGAATAACTGCGTTCCTGCTCAACAT, reverse GTACGACTCACTATAGGGAGGGCAGAATCTTGCACTTTGT;

IKBA forward AGGTGACACTATAGAATACTGCAGCAGACTCCAC, reverse GTACGACTCACTATAGGGAGGGTATTTCCTCGAAAGT;

JNK forward AGGTGACACTATAGAATAAAGGAAAACGTGGATTTATG, reverse GTACGACTCACTATAGGGACCAGCATATTTAGGTCTGTT;

MKP1 forward AGGTGACACTATAGAATAAGAAGAACCAAATACCTCAA, reverse GTACGACTCACTATAGGGACAGGTCATAAATAATCAGCA;

MKI67 forward GCTGAGAACTCCTAAGGGAAAG, reverse GCTGTGAAGCTCTGTAGGATAC;

NFkB forward AGGTGACACTATAGAATACGTTTTAGATACAAATGTGAAG, reverse GTACGACTCACTATAGGGACACTTTTCCTTTTCCATAAT;

NQO1 forward AGGTGACACTATAGAATACTGCGAACTTTCAGTATCC, reverse GTACGACTCACTATAGGGAGAAGGGTCCTTTGTCATAC;

NRF2 forward AGGTGACACTATAGAATATCGCAAACAACTCTTTATCT, reverse GTACGACTCACTATAGGGAAGAGGAGGTCTCCGTTA;

P38 forward AGGTGACACTATAGAATATGAGCTGAAGATTCTGGA, reverse GTACGACTCACTATAGGGATGTCAGACGCATAATCTG;

P53 forward AGGTGACACTATAGAATAATGGAAACTACTTCCTGAAA, reverse GTACGACTCACTATAGGGAATTCTGGGAGCTTCATCT;

PP5 forward AGGTGACACTATAGAATACAAGGACTACGAGAACGCCA, reverse GTACGACTCACTATAGGGAGCTTCACCTTGACCACCGTC;

PP2A forward AGGTGACACTATAGAATACCGCCATTACAGAGAG, reverse GTACGACTCACTATAGGGAAGGATTTCTTTAGCCTTCT;

RPII forward CCCAGCTCCGTTGTACATAAA, reverse TCTAACAGCACAAGTGGAGAAC;

SOD1 forward AGGTGACACTATAGAATAAAGTACAAAGACAGGAAACG, reverse GTACGACTCACTATAGGGATGACAAGTTTAATACCCATCT;

SOD2 forward AGGTAGCATGGACCGAATTTAC, reverse GATAGCCAGGTGTTTGCTTCT.

Each experiment was performed in triplicate and contained 2 wells for each treatment variant. At the qPCR stage, 3 technical repeats were used for each biological sample.

### 4.9. Proliferation Assay via the BrdU Incorporation

The stimulation of the cell proliferation by the ACTH(6–9)PGP peptide was validated using the BrdU cell proliferation kit (Abcam, Cambridge, MA, USA). The cells were seeded in 96-well plates at the density of 4000 per well and grown overnight. After that, peptide solution in the fresh culture medium was added to the cells with full medium replacement; the peptide addition was performed on days 1 and 4 after the seeding. On day 6, BrdU reagent was added to the cells for 24 h, and the assay was completed according to the manufacturer’s protocol.

### 4.10. Statistics

All experiments were performed at least in triplicate. Statistical analysis was performed with the GraphPad Prism 6.0 software using ANOVA with the Holm–Sidak post-test; *p* < 0.05 was considered a statistically significant difference.

## 5. Conclusions

For the first time, we have shown that the peptide combining the ACTH(6–9) fragment and tripeptide PGP (HFRWPGP) protects SH-SY5Y cells against H_2_O_2_, tBH, and KCN cytotoxicity, but not MPP^+^ cytotoxicity. The mechanism of its action was the promotion of proliferation via modulation of proliferation-related *NF-κB* genes and stimulation of prosurvival *NRF2*-gene-related pathway, as well as a decrease in apoptosis.

## Figures and Tables

**Figure 1 molecules-26-01878-f001:**
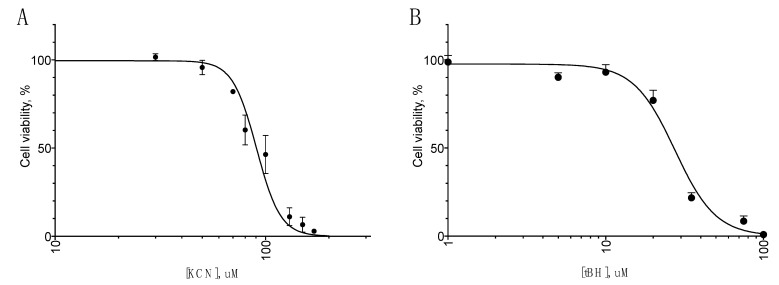
Cytotoxicity of KCN (**A**) and tBH (**B**) for the SH-SY5Y cells. Incubation time 24 h, MTT assay data, mean ± standard error, N = 5 experiments.

**Figure 2 molecules-26-01878-f002:**
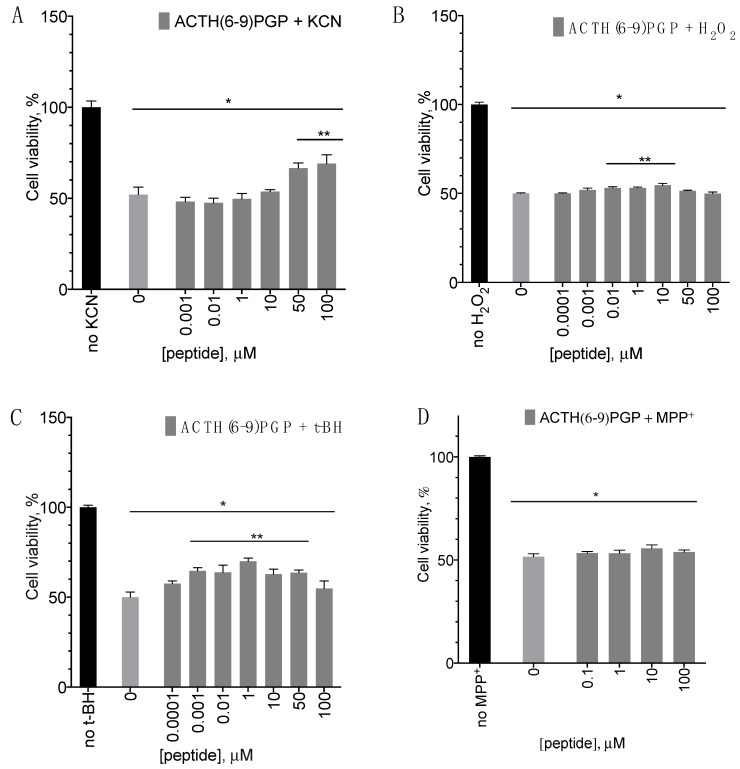
ACTH(6–9)PGP effect on SH-SY5Y cell viability after H_2_O_2_, tBH, MPP^+^, and KCN treatment. The cells were treated with 475 μM H_2_O_2_ (**A**), 27 µM tBH (**B**), 90.6 µM KCN (**C**), or 1.3 mM MPP^+^ (**D**) with various peptide concentrations for 24 h in the case of tBH, H_2_O_2_, and KCN or 48 h in the case of MPP^+^. Untreated cells were used as control. MTT assay data, mean ± standard error, N = 7 experiments. *, a statistically significant difference from the untreated control; **, a statistically significant difference from the control without peptide; ANOVA with the Holm–Sidak post-test, *p* ≤ 0.05.

**Figure 3 molecules-26-01878-f003:**
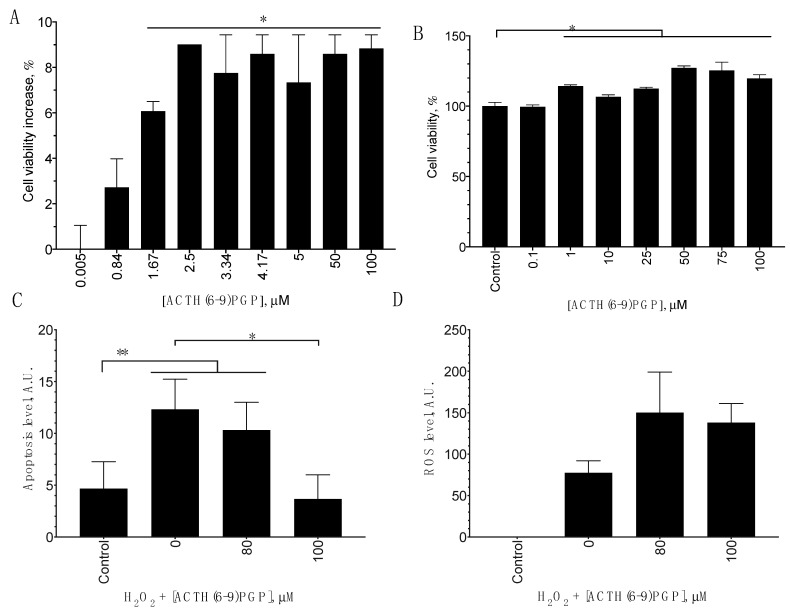
ACTH(6–9)PGP influence on SH-SY5Y cell proliferation (**A**,**B**), H_2_O_2_-induced apoptosis (**C**), and ROS level (**D**). For the proliferation studies, the cells were incubated with the peptide for 7 days and analyzed using the MTT (**A**) or BrdU (**B**) assay. For apoptosis and ROS generation studies, the cells were treated with 475 μM H_2_O_2_ either alone or together with the peptide for 1 h, after which ROS level (using the DCFH-DA dye) and apoptotic cell counts (using the phosphatidylserine-reactive dye combined with the 7-AAD cell-impermeable dye) were determined. Untreated cells were used as control. Mean ± standard error, N = 3 experiments, *, statistically significant difference from H_2_O_2_ alone for ROS and apoptosis and from the untreated control for the proliferation; ANOVA with the Holm–Sidak post-test, *p* ≤ 0.05. **, statistically significant difference from the control without H_2_O_2_, ANOVA with the Holm–Sidak post-test, *p* ≤ 0.05.

**Figure 4 molecules-26-01878-f004:**
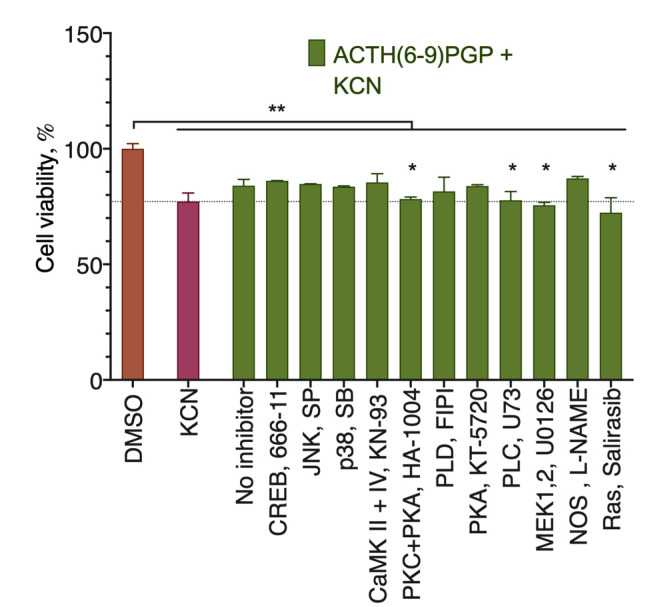
Participation of intracellular signal transduction components in ACTH(6–9)PGP protection against KCN cytotoxicity for the SH-SY5Y cells. Inhibitors for CREB (666-11, 1 µM), JNK (SP 600125, “SP”, 1 µM), p38 (SB 202190, “SB”, 1 µM), CaMKII + IV (KN-93, 4 µM), PKC + PKA (HA-1004, 10 µM), PLD (FIPI, 0.5 µM), PKA (KT-5720, 0.5 µM), PLC (U-73122, “U73”, 10 µM), MEK1/2 (U-0126, 0.2 µM), NOS (L-NAME, 25 µM), and Ras (salirasib, 10 µM) were added 1 h before the KCN and then together with KCN (85.0 µM) and peptide (50 µM). The cells were incubated with the inhibitors, KCN, and peptide for 24 h. MTT assay data, mean ± standard error. *, a statistically significant difference from the KCN + peptide without any inhibitor; **, a statistically significant difference from the untreated control; *p* ≤ 0.05, ANOVA with the Tukey post-test, N = 3 experiments.

**Figure 5 molecules-26-01878-f005:**
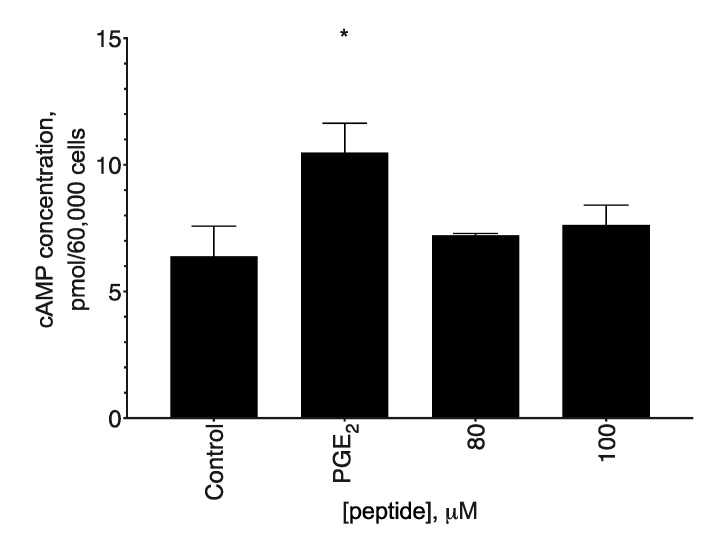
cAMP production after the ACTH(6–9)PGP treatment of SH-SY5Y cells. The cells were treated with 80 or 100 μM of the peptide or with 40 μM of PGE_2_ (positive control) for 20 min, after which the cAMP concentration was measured using a competitive ELISA kit. Untreated cells were used as control. Mean ± standard error, N = 3 experiments. *, a statistically significant difference from the control; ANOVA with the Holm–Sidak post-test, *p* ≤ 0.05.

**Figure 6 molecules-26-01878-f006:**
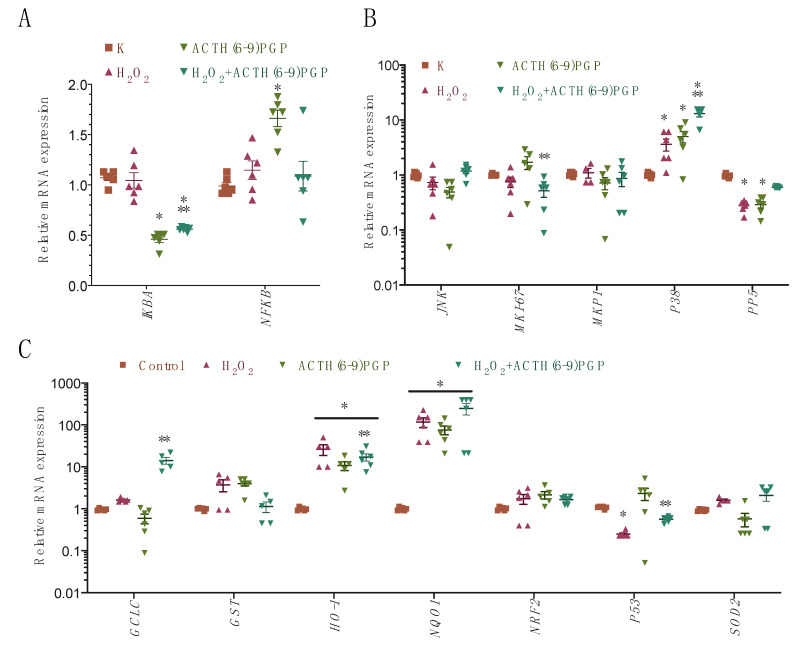
Gene expression changes of NF-κB (**A**), MAPK (**B**), and NRF2 (**C**) pathways after the ACTH(6–9)PGP treatment with and without H_2_O_2_. SH-SY5Y cells were treated with 475 μM of H_2_O_2_ either alone or with 50 μM of the peptide for 24 h, after which mRNA levels were determined using RT-qPCR. Untreated cells were used as control. Data are normalized to *B2M*, *RPII*, and *GPDH*. Mean ± standard error, N = 3 experiments; each data point represents a biological replicate; technical replicates are averaged. *, a statistically significant difference from untreated control; **, a statistically significant difference from the H_2_O_2_ alone; ANOVA with the Holm–Sidak post-test, *p* ≤ 0.05.

## Data Availability

The data presented in this study are available on request from the corresponding author. The data are not publicly available due to legal issues.

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
