# Peer review of "ACTH(6–9)PGP Peptide Protects SH-SY5Y Cells from H_2_O_2_, *tert*-Butyl Hydroperoxide, and Cyanide Cytotoxicity via Stimulation of Proliferation and Induction of Prosurvival-Related Genes"

_molecules, 2021, doi:10.3390/molecules26071878_

Round 1

Reviewer 1 Report

In this manuscript the authors analyzed the neuro-protective effects of  stabilized melanocortin analog peptide ACTH(6-9)PGP on SH-SY5Y stressed with  H2O2, tBH, KCN, and MPP+..  ACTH(6-9)PGP demonstrated a pro-proliferative and anti-apoptotic acting on trascriptional regulation of antioxidant genes.

I found the manuscript interesting and the analysis of  mechanisms implicate on  ACTH(6-9)PGP neuroprotection is of relevance for future studies on  oxidative stress modulation

My comments:

Methods:

Please include the information concerning the “Apoptosis/Necrosis detection kit” used for the experiments. And report what you have quantified in methods and a brief description in the result section.

Please specify the RTqPCR replicates

Figures

The figure and graph are not well presented: different font and size were used for the figures and, I suppose, the “K” reported in the graph should be Control, please fix.

Figures legends : the legends need to be better described and check for grammar, and typo errors.

Results:

Please, fix the nomenclature of genes (all in italic)

From figure 2 the effect of CTH(6-9)PGP on H2O2 is weaker than that on tBH, why did you choose to analyzed only H2O2 toxicity?

Please, present the RtqPCR result as datapoints, it’s preferable .

I suggest to better described the result of the paragraph  “ 2.2. ACTH(6-9)PGP Decreased Apoptosis and Increased Cell Viability, but Did Not Affect Acute 102ROS Level”. In this form is difficult to understand and appreciated the results.

Please, Include references of recent publication, especially when you discuss the antioxidant pathway.

Author Response

Point 1. Methods: Please include the information concerning the “Apoptosis/Necrosis detection kit” used for the experiments. And report what you have quantified in methods and a brief description in the result section.

Response 1. The information on the kit was added to the methods section and to the text.

Point 2. Methods: Please specify the RTqPCR replicates

Response 2. The information was added to the methods section.

Point 3. The figure and graph are not well presented: different font and size were used for the figures and, I suppose, the “K” reported in the graph should be Control, please fix.

Response 3. Figures were corrected.

Point 4. Figures legends: the legends need to be better described and check for grammar, and typo errors.

Response 4. Figure legends were corrected.

Point 5. Results: Please, fix the nomenclature of genes (all in italic)

Response 5. Gene names were converted to italics.

Point 6. Results: From figure 2 the effect of CTH(6-9)PGP on H2O2 is weaker than that on tBH, why did you choose to analyzed only H2O2 toxicity?

Response 6. The effect in the tBH model was more pronounced than in the H2O2 one. However, H2O2 is produced in vivo during oxidative stress and thus represents a more physiological setting. Therefore, further studies on the ACTH(6-9)PGP action mechanism were conducted in the H2O2 model. (The explanation was added to the text).

Point 7. Results: Please, present the RtqPCR result as datapoints, it’s preferable.

Response 7. The format of the figure was changed

Point 8. Results: I suggest to better described the result of the paragraph “ 2.2. ACTH(6-9)PGP Decreased Apoptosis and Increased Cell Viability, but Did Not Affect Acute 102ROS Level”. In this form is difficult to understand and appreciated the results.

Response 8. Paragraph 2.2 was extended to better describe the results

Point 9. Results: Please, include references of recent publication, especially when you discuss the antioxidant pathway.

Response 9. Several references were added to the discussion section

Reviewer 2 Report

The paper "ACTH(6-9)PGP Peptide Protects SH-SY5Y Cells from the H2O2, 2 tert-Butyl Hydroperoxide, and Cyanide Cytotoxicity via Prolif-3 eration Stimulation and Antioxidant-related Genes Induction"is focalized on peptide anologues that possess a wide  range of neuroprotective activities.

Unfourtnately, the peptide design, synthesis and characterization of peptide is lacking. The choice of peptide is not clear and furthermore it is necessary to perform study of peptide stability.

line 50:" They do not cleave the bonds -AA-Pro-, where AA is any amino acid, as well as other sequences enriched in  proline residues" Please add a reference

The results are not so clear:

line 84: EC50 for KCN was 906 µm. Please, write the correct concentration

line 91: "ACTH(6-9)PGP dose-dependently increased cell viability under the treatment with all of the toxic agents except MPP+ (Figure 2A-D). For H2O2 and tBH, the protective ac- 92 activity maximum war observed at the concentration around 1-10 μM (Figure 2 A, B)" This conclusion is not true. In particular under the treatment with H2O2 there is not a dose dependent cell viability  as for cells treated with MPP+. 

Moreover, for a correct interpretation of the paper  is necessary report the data above the cytotoxicity of H2O2 and MPP enen if previously determined

Figure 3: increase the quality of figure. Report the peptide concentration and the label are not visible

Moreover, the discussion is not focused

Author Response

Point 1. Unfortunately, the peptide design, synthesis and characterization of peptide is lacking. The choice of peptide is not clear and furthermore it is necessary to perform study of peptide stability.

Response 1. The peptide was synthesized by methods of classical peptide chemistry in solution using both protected and free L-amino acids, as described earlier [Shevchenko, K. V.; Dulov, S.A.; Andreeva, L.A.; Nagaev, I.Y.; Shevchenko, V.P.; Radilov, A.S.; Myasoedov, N.F. Stability of His-Phe-Arg-Trp-Pro-Gly-Pro to Leucine Aminopeptidase, Carboxypeptidase Y, and Rat Nasal Mucus, Blood, and Plasma. Russian Journal of Bioorganic Chemistry 2016, 42, 153–161, doi:10.1134/S1068162016020126.]. Purity of the final compound was not less 98% (HPLC analysis, Supplement S2).  final peptide characteristics were added to the Supplement 2. The corresponding text was incorporated within the manuscript. Rationale for peptide structure choice was explained in Introductory section and some key references were added.  As for stability of PGP flanked peptides, it was investigated well for approved drug Semax. Duration of physiological effect of Semax in T-shape maze educational test was 50 times as natural ACTH(4-10) peptide. Stability of ACTH(6-9)PGP was also recently studied using several peptidases and it was shown that under all experimental conditions WG-PGP bond was resistant to hydrolytic splitting. The corresponding text and references were added to the Introduction section.

Point 2. line 50:" They do not cleave the bonds -AA-Pro-, where AA is any amino acid, as well as other sequences enriched in proline residues" Please add a reference

Response 2. The pioneering work of Walker and al. [Walter, R.; Simmons, WilliamH.; Yoshimoto, T. Proline Specific Endo- and Exopeptidases. Mol Cell Biochem 1980, 30, doi:10.1007/BF00227927] was added.

Point 3. line 84: EC50 for KCN was 906 µm. Please, write the correct concentration

Response 3. The typo in the concentration decimal point was corrected.

Point 4. line 91: "ACTH(6-9)PGP dose-dependently increased cell viability under the treatment with all of the toxic agents except MPP+ (Figure 2A-D). For H2O2 and tBH, the protective ac- 92 activity maximum war observed at the concentration around 1-10 μM (Figure 2 A, B)" This conclusion is not true. In particular under the treatment with H2O2 there is not a dose dependent cell viability as for cells treated with MPP+. 

Response 4. The description was corrected.

Point 5. Moreover, for a correct interpretation of the paper is necessary report the data above the cytotoxicity of H2O2 and MPP even if previously determined

Response 5. The EC50 values for H2O2 and MPP+ were added to the text

Point 6. Figure 3: increase the quality of figure. Report the peptide concentration and the label are not visible

Response 6. Figure quality was improved.

Point 7. Moreover, the discussion is not focused

Response 7. The comparison of the peptide to other known antioxidant and neuroprotective peptides was elaborated. The explanation of the choice of the ACTH(6-9)PGP peptide was added at the beginning of the Discussion section.

Round 2

Reviewer 2 Report

Accept in presence form